# MULTI-AGENT LANGUAGE LEARNING: SYMBOLIC MAPPING

## ABSTRACT

The study of emergent communication has long been devoted to coax neural network agents to learn a language sharing similar properties with human language. In this paper, we try to find a 'natural' way to help agents learn a *compositional* and *symmetric* language in complex settings like dialog games. Inspired by the theory that human language was originated from simple interactions, we hypothesize that language may evolve from simple tasks to difficult tasks. We propose a novel architecture called *symbolic mapping* as a basic component of the communication system of agent. We find that symbolic mapping learned in simple referential games can notably promote language learning in difficult tasks. Further, we explore *vocabulary expansion*, and show that with the help of symbolic mapping, agents can easily learn to use new symbols when the environment becomes more complex. All in all, we probe into how symbolic mapping helps language learning and find that a process from simplicity to complexity can serve as a natural way to help multi-agent language learning.

## 1 INTRODUCTION

Agent communication has been a popular research field in the context of multi-agent reinforcement learning (Foerster et al., 2016; Sukhbaatar et al., 2016; Jiang & Lu, 2018; Eccles et al., 2019). Recent work has focused on the emergence of language in cooperative tasks where neural network agents learn a communication protocol from scratch to solve problems together (Lazaridou et al., 2017; Das et al., 2017; Havrylov & Titov, 2017; Kottur et al., 2017; Li & Bowling, 2019; Ren et al., 2020). An array of work has empirically shown that agents can make use of their developed language to successfully complete the tasks. Beyond that, some work probed into the process of language emergence, and tried to figure out whether the learned language could share similar properties with human language like *compositionality* (Mordatch & Abbeel, 2018; Resnick et al., 2020; Chaabouni et al., 2020; Choi et al., 2018) and *symmetry* (Graesser et al., 2019; Dubova & Moskvichev, 2020; Dubova et al., 2020).

Most of these studies on emergent communication are based on *referential games* (Lewis, 1969) and have shown that compositionality can be induced by adding suitable environmental pressures. Some have explored the influential factors on symmetry of communication protocols among a group of agents. However, tasks in these studies are often simple, and some of these methods are hard to implement in complex settings like dialog games. Kottur et al. (2017) found that in a two-agent multi-round dialog game, language with compositionality does not naturally emerge, unless strict conditions are imposed to agents, such as deprivation of memory.

Language emergence only in simple tasks is obviously not satisfactory. In this paper, we tend to find a new way to make compositional and symmetric language emerge 'naturally' in complex settings. Psychological studies suggest that human language was originated from simple gestures like pointing and pantomiming (Tomasello, 2010). This may explain why simple referential games are suitable for emergent language studies: these games are similar to 'pointing' in pragmatic process. However, from another perspective, the theory may also imply that communication protocols like human language cannot be formed *directly* from complex interactions. Instead, a natural process is probably that a language is first formed in simple tasks, and then applied in more complex tasks, meanwhile it evolves to become more complicated and complete, similar to the concept of *curriculum learning* (Bengio et al., 2009). This is reasonable because a well-structured communication protocol is hard

to form, and a complex setting or a difficult task exacerbates the problem. On the other hand, an important characteristic of language is that it is a general capacity. Once learned, it should be helpful in all kinds of tasks that need communication. Hence, we design two games, including a two-player referential game and a multi-round dialog game involving a group of agents, and investigate whether the same trend also arises on the communication protocol learned by neural network agents, so that language could evolve from simple tasks to difficult tasks and this process, which we call as *task transfer*, could help language learning in complex settings.

In order to implement this process, we need find a way to enable agents to learn a common function for communication, because the speaking and listening policy can be different across tasks and thus should not be transferred directly. We propose a novel architecture called *symbolic mapping*, which maps the input to related symbols, as a basic component of communication system of agent. The intuition is that when presented with the same input, we always associate it with the same pile of words and concepts, and this kind of association is consistent across tasks. This association does not determine the communication protocol, but it can encode language properties and be shared for communication at all time. Our experiments show that by implementing symbolic mapping, agents can achieve higher success rate in difficult tasks after training in simple referential games, and the learned communication protocol presents better language properties.

As we explore the learning process of agents from simple tasks to difficult tasks, we are also curious about how the language becomes more complicated when old conventions are not enough in new environments. Language learning should not be accomplished overnight, and sometimes agents cannot learn a language well if the environment is complex at the beginning. In a more natural scene, agents should first learn a simple language in a simple initial environment, and when they enter a more complicated environment, they will learn something new and the language develops. We conduct the experiment about *vocabulary expansion*, and find that agents with symbolic mapping can learn to communicate using new symbols well in new environments. And through vocabulary expansion agents can accomplish tasks in complex environments where they would fail if they are asked to learn a language directly. This result once again reveals that a process from simplicity to complexity is crucial for multi-agent language learning.

## 2 RELATED WORK

**Cooperative games.** Different kinds of cooperative games have been proposed in emergent communication literature. A popular one is referential game (Lewis, 1969), where one agent, often noted as the speaker, has to send a message describing a target (*e.g.*, an image) which it has just observed to the other agent. Then the other agent, often noted as the listener, must select the target from several candidates containing the target and some distractors, after receiving the message (Lazaridou et al., 2017; Havrylov & Titov, 2017). We design a variant of referential game to serve as the simple task in our experiments. The game in Chaabouni et al. (2020) is most similar to our simple task, where the listener should reconstruct the target instead of picking out the target from a pool of candidates. The main difference is that in our task, the listener model is trained by reinforcement learning, while they use the cross-entropy loss to train the listener.

Our difficult task is inspired by the *Task & Talk* game proposed by Kottur et al. (2017), which is a multi-round dialog game. In the Task & Talk game, there are two agents, one always asks questions while the other always answers questions. However, our task involves a group of homogeneous agents who do not play specific roles in the game. Besides, agents in our task can choose to end the dialog any time before the number of dialog rounds reaches the upper limit. But in the Task & Talk game, the number of dialog rounds is fixed. Other studies (Mordatch & Abbeel, 2018; Graesser et al., 2019; Fitzgerald, 2019) also concern emergent language in a group of agents, and Evtimova et al. (2018) proposed a multi-step referential game. However, no game in these studies is similar to ours.

**Properties of communication protocols.** A mainstream research direction in emergent communication is to find out whether neural network agents can produce communication protocols which exhibit some properties of human language. The most extensively studied property is compositionality. Many studies (Lazaridou et al., 2018; Li & Bowling, 2019; Ren et al., 2020; Resnick et al., 2020) have found that in referential games, once given appropriate environmental pressures, like changing learning environments, communication capacities or agents' model capacities, compositionality could be improved. Kottur et al. (2017) found that compositionality does not emerge naturally in dialog

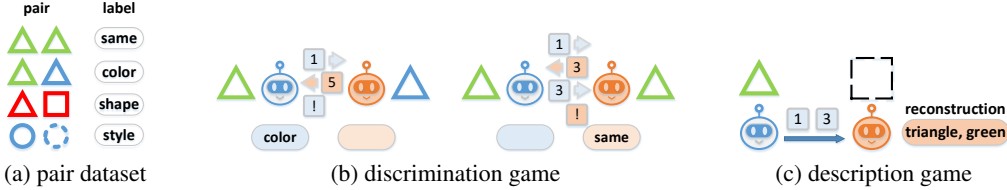

Figure 1: Dataset and games.

games, which is also verified by our experiment. In the studies where a group of agents learn their languages together, another important communication property is symmetry. That means an agent community should converge on a shared communication protocol. Dubova et al. (2020) investigated the impact of different social network structures on language symmetry, while Dubova & Moskvichev (2020) explored some other factors including supervision, population size and self-play. In this paper, we focus on improving the two properties through a process from simplicity to complexity, and we propose an architecture called *symbolic mapping* to implement the process.

**Evolution of communication.** Recent studies, inspired by linguistic theories, have brought evolution into the research of emergent communication. Cogswell et al. (2019) investigated the benefit from cultural transmission, while Dagan et al. (2021) integrated both cultural evolution and genetic evolution into emergent communication. Ren et al. (2020) proposed a neural iterated learning algorithm, where agents in a new generation are partially exposed to the language emerged from the previous generation. Li & Bowling (2019) let the speaker interact with new listeners periodically, while Graesser et al. (2019) analyzed how the learned language evolves when different linguistic communities come in contact with each other. Most similar to our approach, Korbak et al. (2019) explored language learning across games of varying complexity by template transfer. Different from their work where a hard task is decomposed into several parts and the transferred agent is the listener, we explore language transfer from simple interactions to different tasks involving more complex communication forms, and the speaker is not reinitialized so that the language evolution is consistent. And we also explore the expansion of vocabulary.

**Symbolic representation.** Previous studies have explored symbolic representation in the deep reinforcement learning (RL) framework (Garnelo et al., 2016; Garnelo & Shanahan, 2019), and found that a compositionally-structured representation could help address several shortcomings inherent in the deep RL systems. Symbolic mapping can be seen as a kind of symbolic representation in its function. Different from prior work, symbolic mapping is learned and constructed through emergent communication instead of representation learning techniques and is trained end-to-end by RL. That means agents form the symbolic representation when learning to communicate.

## 3 EXPERIMENTAL FRAMEWORK

### 3.1 GAME SETTINGS

**Discrimination game.** We explore emergent communication in the context of a multi-round dialog game as the difficult game, illustrated in Figure 1b. Our game includes a group of homogeneous agents, which we call a community. Each game episode involves two agents $i$ and $j$ which are randomly sampled from the community. They are presented with object $o_i$ and $o_j$ respectively. The object pair $(o_i, o_j)$ is sampled from a pair dataset $\mathcal{P}$. Each pair in $\mathcal{P}$ contains two objects selected from an object dataset $\mathcal{D}$. Each object in $\mathcal{D}$ comprises $n$ attributes. For each attribute $a \in \{1, 2, \ldots, n\}$, there are $m^{(a)}$ possible values. For a given $n$ and a tuple of value numbers $m = (m^{(1)}, m^{(2)}, \ldots, m^{(n)})$, we note the corresponding object dataset as $\mathcal{D}_{n,m}$, and the number of different objects will be $|\mathcal{D}_{n,m}| = \prod_{a=1}^{n} m^{(a)}$. Given an object dataset $\mathcal{D}$, the pair dataset $\mathcal{P}$, as illustrated in Figure 1a, is then constructed as for each pair $(o_i, o_j)$ where $o_i, o_j \in \mathcal{D}$, $o_i = o_j$ or $o_i$ and $o_j$ have only one different attribute. If the objects are selected from $\mathcal{D}_{n,m}$, we note the pair dataset as $\mathcal{P}_{n,m}$. Note that different orders of $o_i$ and $o_j$ mean different pairs, since $o_i$ will be observed by agent $i$ who will speak first in a game episode. Moreover, each pair $p = (o_i, o_j) \in \mathcal{P}$ has a label $l_p$. If $o_i = o_j$, then $l_p = 0$; otherwise if $o_i$ and $o_j$ are different in attribute $a$, then $l_p = a$.

After observing their respective objects, two agents start the dialog. At each time step $t$, the speaking agent should choose a symbol $s_t$ from a shared vocabulary $V$ and send it to the other agent. Any

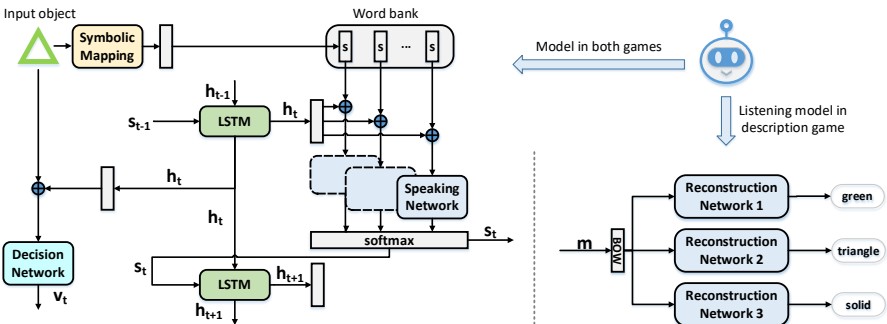

Figure 2: Agent architecture.

agent, after receiving a symbol, can choose to continue or terminate the dialog. If the choice is to continue, then the receiving agent becomes the speaker at the next time step, and the players take turns to speak until the dialog is terminated. Suppose agent $j$ chooses to end the dialog, then it must answer whether $o_i$ and $o_j$ are the same; if not, then which attribute is the different one. In other words, it must pick the true label $l_p$ for the pair $(o_i, o_j)$. If the answer is correct, then both agents succeed and get a reward $r = 1$. Otherwise, they fail and get no reward ($r = 0$). If the number of dialog rounds reaches the upper limit $T_{\max}$, the agents also fail.

**Description game.** We also design a variant of referential game called description game, as depicted in Figure 1c. The game proceeds as follows. First, an agent $i$ receives an input object $o_i$ from $\mathcal{D}_{n,m}$. Next, it chooses a fixed-length ($n$) sequence of symbols from vocabulary $V$ to describe $o_i$, and sends it to listener $j$. After that, $j$ consumes all symbols and outputs $\hat{o}_i$. If $o_i = \hat{o}_i$, which means $j$ successfully reconstructs what agent $i$ is talking about, the agents succeed. Otherwise, they fail. The reward for speaker $i$ is according to the game result, namely $r = 1$ if they succeed or $r = 0$ if they fail. The reward for listener $j$ is according to its reconstruction of each attribute. In our setting, the listener has a reconstruction model for each attribute. Each reconstruction model gets $r = 1$ if its corresponding attribute is reconstructed correctly, and gets $r = 0$ otherwise.

### 3.2 AGENT ARCHITECTURE

We propose a novel agent architecture which contains symbolic mapping, shown in Figure 2. First we illustrate how symbolic mapping works. Concretely, for each attribute $a$, we represent it as a $N_a$-dimension one-hot vector, where $N_a = m^{(a)}$. An input object $o_i$ from $\mathcal{D}_{n,m}$ is represented by the concatenation of all its attributes. Symbolic mapping $\text{map}(\cdot)$, realized by a linear layer followed by a sigmoid function, maps the input object $o_i$ to a vector with dimension $|V|$, and each element of the vector corresponds to the degree of relevance between a symbol and the object. Several symbols are sampled using the Bernoulli distribution for each element of the vector according to the probability given by the output of the sigmoid function, and then stored as the agent's *word bank*. The number of sampled symbols, namely the size of the word bank, is not predefined or limited.

In discrimination game, an LSTM $f(\cdot)$ serves as an agent's memory. We initialize the hidden state $h_0$ as a zero vector, and each time a symbol $s$ is transmitted in the dialog, the symbol $s$ is fed into $f(\cdot)$. We encode symbols into one-hot embeddings. To differentiate the speaker of each symbol, we concatenate a flag to the embeddings. If the speaker is the agent itself, the flag is zero; if the speaker is another agent, the flag is one. Note that the agent does not know the *identity* of its partner. Suppose at time $t$, agent $i$ is ready to speak, and agent $j$ is its partner. Each symbol $s$ in the word bank of agent $i$ will be encoded into a one-hot embedding. Then each embedding is concatenated to the agent's hidden state $h_t^i$, and the speaking network $g_{\text{sp}}^i(\cdot)$, realized by a 2-layer MLP, takes each of them as input and outputs a score for each symbol $s$. All symbols in the word bank get scores by the shared speaking network $g_{\text{sp}}^i(\cdot)$, and all scores will be passed through a softmax function to get a probability distribution $\pi_{\text{sp}}^i(\cdot)$ over the word bank $w_i$, and a symbol $s_t$ will be sampled from $\pi_{\text{sp}}^i(\cdot)$. The symbol $s_t$ is then fed into both agents' LSTM $f^i(\cdot)$ and $f^j(\cdot)$. At the next time step $t + 1$, agent $j$ passes the concatenation of its hidden state $h_{t+1}^j$ and input object $o_j$ into a decision network $\pi_{\text{dec}}^j(\cdot)$, realized by a 2-layer MLP, and outputs an action $v_{t+1}$. If the action is to continue, then it is time for agent $j$ to speak. Otherwise, the action means the answer, and both agents get the corresponding reward.

In description game, the architecture of the speaker agent is the same as above, except that the decision network is not used. We fix the message length to $n$, corresponding to one symbol per attribute. To do this, after the speaker produces a symbol $s_t$ at time step $t$, the symbol is fed into its memory network $f(\cdot)$, and the next symbol $s_{t+1}$ is sampled at time step $t+1$. This process proceeds until the fixed message length is reached. The listener is instantiated by $n$ linear layers, which are called reconstruction networks. The message sent by the speaker is represented by the *bag-of-words model* and consumed by the listener. Then each of its reconstruction network outputs an action to predict the value of each attribute of the object.

In both games, we use REINFORCE (Williams, 1992) to train each agent end to end. We apply entropy regularization in the loss function to encourage exploration, and use the Adam optimizer with learning rate 0.001 in all settings. We run all our experiments three times with different random seeds and present the mean and standard deviation of the results.

### 3.3 THE BENEFIT OF SYMBOLIC MAPPING

In this section we illustrate the benefit of symbolic mapping briefly. One advantage brought by symbolic mapping is the capability to easily transfer between different tasks. While neural network has limited generalising capabilities to new tasks, the symbolic association should represent more abstract concepts which can help task transfer. Besides, though the mapping is simple, it can encode the basic component of a language, so it can help maintain language properties across tasks. For example, a compositional structure of the symbolic mapping can help maintain compositionality. Another benefit is that symbolic mapping is suitable for vocabulary expansion.Since our speaking network chooses symbols from the word bank rather than sampling from a fixed-dimension distribution, we can explore vocabulary expansion just by adding outputs to symbolic mapping so that agents can associate input objects with more symbols. We will verify these benefits in our experiments.

### 3.4 METRICS

**Compositionality.** In our setting, the evaluation criterion of compositionality is whether agents can communicate different attributes independently. Note that compositionalty in natural language has more complicated forms, but we only consider the juxtaposition of independent symbols to represent an overall meaning because we hypothesize that compositionality was rather simple when language was formed in the early stage and thus the proposed form is adequate for our research. Inspired by *positional disentanglement* in Chaabouni et al. (2020), we propose a metric called **referential disentanglement** (*refdis*), which measures whether a specific symbol refers to a specific attribute. We ignore the positional information because we need a language suitable for different kinds of interactions, and if symbols' positions are informative, the language is hard to transfer to dialogs.

For each symbol $s$ in the vocabulary, we denote $a_1^s$ the attribute that has the lowest conditional entropy given $s$ : $a_1^s = \arg\min_a \mathcal{H}(a|s)$. Similarly, we denote $a_2^s = \arg\min_{a \neq a_1^s} \mathcal{H}(a|s)$, which has the second lowest conditional entropy. Then we define *refdis* as:

$$refdis = \sum_s \left( \frac{\mathcal{H}(a_2^s|s)}{\mathcal{H}(a_2^s)} - \frac{\mathcal{H}(a_1^s|s)}{\mathcal{H}(a_1^s)} \right) \cdot k(s), \tag{1}$$

where $k(s)$ is the frequency of occurrence of symbol $s$. The intuition of equation 1 is that each symbol should only be informative about one attribute. The best case is when one attribute is determined but all other attributes are totally uncertain given any specific symbol, with *refdis* being 1, and in the worst case the *refdis* is 0. *Context-independence* (CI) proposed in Bogin et al. (2018) shares similar concept with *refdis*, but *refdis* evaluates compositionality according to symbols while CI focuses on the alignment between symbols and concepts.

**Symmetry.** We evaluate the symmetry of the learned communication protocol by computing the Jensen-Shannon divergence between pairs of agents' distributions of different values of attributes, given a specific symbol. For a pair of agents $i$ and $j$, we define **referential divergence** (*refdiv*) as:

$$refdiv = \frac{1}{|V| \cdot n} \sum_s \sum_a \text{JSD} \left( p(m_i^a|a,s) \| p(m_j^a|a,s) \right), \tag{2}$$

where $p(m_i^a|a,s)$ is the value distribution of attribute $a$ of agent $i$ given symbol $s$. The value of *refdiv* is also between 0 and 1, and a perfectly symmetric communication protocol will get *refdiv* = 0.

Table 1: The performance of the agent community playing discrimination game on dataset $\mathcal{P}_{3,(3,3,3)}$. LSTM refers to vanilla LSTM-based agents, while IL refers to LSTM agents trained with iterated learning. The first and second column shows the success rate in training set and testing set respectively.

| | Training (%) | Testing (%) | *refdis* $\uparrow$ | *refdiv* $\downarrow$ |
|---|---|---|---|---|
| LSTM | 47.62(2.54) | 8.42(1.27) | 0.07(0.03) | 0.87(0.11) |
| IL | 45.67(0.66) | 13.47(1.27) | 0.06(0.01) | 0.87(0.02) |

## 4 EXPERIMENTS

### 4.1 LANGUAGE LEARNING IN DISCRIMINATION GAME

We first examine the performance of neural network agents leaning language in discrimination game, which is a multi-round dialog game involving a group of agents, and the round number is not fixed. We test two methods: vanilla LSTM, which is aimed to show the performance of simple LSTM-based agents in the game without particular training methods, and iterated learning (IL), which is a framework proposed by evolutionary linguists to simulate the language evolution process, and is believed to help compositional languages emerge (Kirby et al., 2014). To apply iterated learning in our setup, we modify the neural iterated learning algorithm (NIL) proposed by Ren et al. (2020). The implementation details of LSTM and IL can be found in appendix. We use dataset $\mathcal{P}_{3,(3,3,3)}$, and refer to the attributes as *color*, *shape* and *style*, and each of them has 3 values (*i.e.*, red, green, blue, triangle, square, circle, solid, dotted, filled). We also split the dataset into the training set and the testing set to explore the generalization ability of the learned languages, which can also reflect the compositionality. We set agent number to 3, and the vocabulary size is set to 9. The upper limit for the number of dialog rounds is $T_{\max} = 3$ (each agent has three turns to speak).

Table 1 shows the results, where *refdiv* is averaged over all pairs of agents. We can find that both two methods get poor performance in discrimination game. The success rates reveal that agents encounter difficulties in learning a good policy to accomplish the game, and their learned communication protocols are overfitting the training set, which implies that the language is not compositional. The low *refdis* also verifies this. Besides, the results of *refdiv* show that the agents do not converge on symmetric communication protocols. These results confirm that the multi-round dialog game is challenging for a good language to emerge. And methods like iterated learning may not work well in complex settings, though the IL agents achieve relatively higher success rate in the testing set.

We conjecture that the difficulty may come from the following reasons. For compositionality, agents need to express a complex object with multiple symbols each referring to a component element. In a speaker-listener game, agents are free to send all relevant symbols at a time, and the structure of language can be determined by themselves. But in a dialog, an agent cannot predict what its partner will say in the next round, and when the dialog will be terminated. On one hand, the language structure must be flexible enough to respond to different coming messages, which is harder to be explored. On the other hand, this kind of instability may push the agents to convey more information each time (*e.g.*, using one symbol to express both two attributes), ending up in a non-compositional communication protocol. For language symmetry, in an agent group, different partners may decode a same message in different ways, and as a result the communication will be unstable and hard to converge on a shared communication protocol. On the other hand, agents cannot get high success rate if they speak differently, so the demand to converge on a symmetric language makes language learning more difficult. Therefore, learning language directly in discrimination game is hard. However, as aforementioned, a natural process is probably that a language is first formed in simple tasks, and we should not train agents in the complex settings to learn a language from scratch.

### 4.2 FROM SIMPLE TASKS TO DIFFICULT TASKS

In this section, we want to verify our hypothesis that language can evolve from simple tasks to difficult tasks, and this process, which we call as *task transfer*, helps language learning in difficult tasks. To do this, we first carry out description game on the agent community, and then train the learned speakers to play discrimination game. And we want to investigate whether our proposed symbolic mapping architecture can indeed promote this process, so we use LSTM and IL introduced in the previous section to serve as our baselines.

Table 2: The performance of the agent community playing with a shared listener in description game on dataset $\mathcal{D}_{3,(3,3,3)}$. SM refers to agents with the proposed architecture. The two metrics are calculated on both symbolic mapping and communication protocol for SM agents.

| | | Success Rate (%) | refdis ↑ | refdiv ↓ |
|---|---|---|---|---|
| LSTM | | 100.00(0.00) | 0.48(0.07) | 0.06(0.04) |
| IL | | 100.00(0.00) | 0.71(0.09) | 0.19(0.03) |
| SM | protocol | 100.00(0.00) | **0.89**(0.06) | 0.12(0.03) |
| | mapping | | 0.71(0.20) | 0.04(0.04) |

Table 3: The performance of the agent community playing discrimination game after they have learned to accomplish description game.

| | | Training(%) | Testing(%) | refdis ↑ | refdiv ↓ |
|---|---|---|---|---|---|
| LSTM | | 85.80(2.82) | 51.01(10.14) | 0.34(0.05) | 0.28(0.08) |
| IL | | 51.13(4.87) | 15.66(5.82) | 0.05(0.03) | 0.75(0.09) |
| SM | protocol | **94.17**(4.98) | **85.35**(8.27) | **0.62**(0.08) | 0.18(0.06) |
| | mapping | | | 0.37(0.09) | **0.06**(0.01) |

### 4.2.1 LANGUAGE LEARNING IN DESCRIPTION GAME

To conduct a speaker-listener game in an agent community, most studies make each agent both speaker and listener to simulate a human community (Dubova & Moskvichev, 2020; Dubova et al., 2020). However, we argue that agent community performs differently from human community so that this way makes language learning more difficult. When human beings learn expressions from each other, they tend to imitate them and speak in the same way (Garrod & Doherty, 1994), so their speaking and listening are tied together. But for neural network agents, it is a different story since their speaking policy and listening policy are separate from each other. Concretely, after agent $j$ learns from agent $i$ by listening, it may speak to agent $k$ in another way. From this perspective, the setting where each agent is both speaker and listener can be seen as multiple speaking models speaking to multiple listening models, making the learning unstable and hard to converge.

Therefore, instead of giving each agent a listening model to interact with all other agents, we choose to use a *shared listener* to simplify and stabilize the language learning. And as mentioned above, different partners decoding a same message in different ways makes symmetric language hard to be learned, but the shared listener can solve this problem. As we need each agent in the agent community play the speaker role to train the speaking policy, we additionally introduce another agent to play the listener role.

We use dataset $\mathcal{D}_{3,(3,3,3)}$, and set agent number in the community to 3 and vocabulary size to 9, the same as in Section 4.1. The message length is set to 3. The results are shown in Table 2. SM refers to agents with the proposed architecture in Section 3.2, and for SM agents we calculate the two metrics on both symbolic mapping (which symbols are stored into word bank) and the actual communication protocol (which words are sent to another agent) to explore their relationship. All methods can learn to accomplish the game perfectly, and results of *refdiv* show that agents can converge on symmetric languages more easily in this simple referential game. Besides, the languages that emerge in this game present much higher compositionality compared with language learned in discrimination game, confirming that simple tasks are more suitable for agents to learn language with good properties.

Among the three methods, LSTM agents achieve relatively poor compositionality, showing that agents cannot learn compositionality so well without any environmental pressure, in line with conclusions in other studies. IL agents perform much better in terms of compositionalty, so the method can indeed help in this simpler game. The relatively poor symmetry may be caused by the supervised learning phase in iterated learning, where each new agent learns language from different agents in the past generation. Languages learned by SM agents present best compositionality. This may be because that the symbolic mapping naturally promotes compositionality, since the association between input and symbols can be easily disentangled. High *refdis* and low *refdiv* calculated on symbolic mapping also indicate that after language learning, the mapping can encode good language properties.

### 4.2.2 TASK TRANSFER

After the agents have successfully learned to accomplish description game, we then train the speakers to play discrimination game. For LSTM agents, we use the learned model directly in the new task. For IL agents, we use the learned model to perform task transfer in the first generation. For SM agents, we load the learned symbolic mapping to reinitialized models, and we do not fix the symbolic mapping so that it can continue to evolve. The experiment settings are the same as in Section 4.1.

The results are shown in Table 3. The performance improvement of LSTM and IL compared with that in Table 1 proves our hypothesis that making languages formed through simple interactions first and then applied in more complex tasks can be a natural way for agents to learn good language in complex settings. Further, the best performance of SM agents confirms the benefit of our proposed architecture. In different kinds of games, agents need different speaking policies, so LSTM and IL agents, who transfer the speaking policies directly, cannot generalize so well to the new game. IL agents perform relatively bad in task transfer probably because that in the last few generations when training in the simple game, they reinforce the successful policy again and again, and they learn the policy for the simple game so firmly that the generalization to a new task becomes more difficult.

In contrast, SM agents learn a new speaking policy from scratch in the new game, while the symbolic mapping provides knowledge about the learned language implicitly. The association from input to symbols encodes the information and properties of a language which is not tied to specific tasks, so SM agents can transfer the language more easily, and maintain the properties like compositionality and symmetry better. We note that *refdis* calculated on symbolic mapping here is relatively low. The reason is that there are several redundant symbols associated, which may reflect that the symbolic mapping becomes more conservative in the difficult game. However, the speaking policy is not constrained by it, and SM agents can make use of the symbolic mapping to find a compositional language.

## 4.3 VOCABULARY EXPANSION

We have empirically shown that our agents' language can evolve in task transfer, and in this section we explore whether language can evolve when the environment becomes more complex. In natural language, it is common that vocabulary changes continually over time and new words are created endlessly, so we hope language emerged by agents can also develop. Besides, the emergence of language should not be accomplished overnight, and a natural process is to form the language step by step.

We explore this question by conducting the experiment called *vocabulary expansion*. We first carry out description game using LSTM and SM agents on dataset $\mathcal{D}_{3,(4,4,4)}$ which contains 64 objects. We set agent number to 3 and vocabulary size to 12. The results are shown in Table 4. It is surprising that in this bigger dataset, both methods fail in the simple task. LSTM agents learn only to speak a single word all the time, and the symbolic mapping learned by SM agents is nearly random. The reason is probably that when the object number in the environment is big, the chance to succeed is very small at the beginning, *e.g.,* $1/64$ in this setting. Then the reward is very sparse and reinforcement agents will find it hard to learn.

Now we try to make agents learn the language from a simpler start. We first train the agents on a smaller dataset $\mathcal{D}_{2,(4,4)}$, and then we introduce a new attribute into the environment and train them on $\mathcal{D}_{3,(4,4,4)}$. This simulates a setting where agents do not care about objects' styles at first so they only learn to communicate about colors and shapes, but when they find that there are also many kinds of styles of objects, they start to learn to communicate about them too.

When training the description game on $\mathcal{D}_{2,(4,4)}$, we use zero-padding to object representations and symbol embeddings to encode the new attribute and new symbols, and we set message length to 2. The vocabulary size is set to 8 at first. For LSTM agents, the output number of the speaker network is set to 12, but we mask 4 of them in the first training. When training the three attribute game, the message length is added to 3, and the vocabulary size is expanded to 12. We use the learned model directly for LSTM agents. For SM agents, we reinitialize the agents' symbolic mapping as a linear layer with output dimension $dim = 12$ and set the weights to be zero. Then we load the parameters of the learned symbolic mapping into it. We also try to reinitialize the speaker network and the LSTM

Table 4: The performance of the agent community playing with a shared listener in description game on dataset $\mathcal{D}_{3,(4,4,4)}$.

| | | Success Rate (%) | refdis ↑ | refdiv ↓ |
|---|---|---|---|---|
| LSTM | | 1.56(0.00) | 0.00(0.00) | 1.00(0.00) |
| SM | protocol mapping | 2.77(0.60) | 0.09(0.06) | 0.75(0.07) |
| | | | 0.03(0.01) | 0.18(0.07) |

Table 5: The performance of the agent community playing with a shared listener in description game on dataset $\mathcal{D}_{2,(4,4)}$.

| | | Success Rate (%) | refdis ↑ | refdiv ↓ |
|---|---|---|---|---|
| LSTM | | 100.00(0.00) | 0.64(0.12) | 0.11(0.06) |
| SM | protocol mapping | 100.00(0.00) | 0.84(0.06) | 0.12(0.02) |
| | | | 0.59(0.18) | 0.05(0.01) |

Table 6: The performance of the agent community playing with a shared listener in description game on dataset $\mathcal{D}_{3,(4,4,4)}$ after vocabulary expansion. SM-reinitialized means the speaker network and the LSTM network of SM agents are reinitialized.

| | | Success Rate (%) | refdis ↑ | refdiv ↓ |
|---|---|---|---|---|
| LSTM | | 83.85(22.65) | 0.47(0.25) | 0.14(0.05) |
| SM | protocol mapping | 100.00(0.00) | **0.91**(0.03) | 0.11(0.02) |
| | | | 0.73(0.10) | 0.05(0.01) |
| SM-reinitialized | protocol mapping | 100.00(0.00) | **0.91**(0.01) | 0.12(0.04) |
| | | | 0.72(0.04) | 0.06(0.02) |

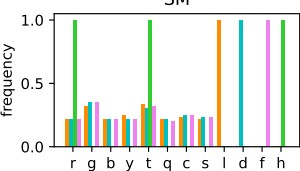
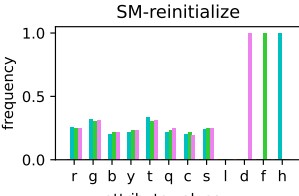
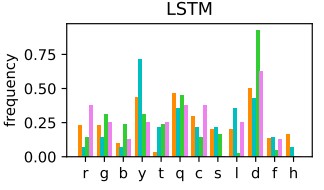

Figure 3: The frequencies of attribute values observed by LSTM and SM agents corresponding to four new symbols in the vocabulary expansion experiment. The four colors of bars correspond to four new symbols respectively. The *x* label is abbreviations of attribute values, and the last four are values of the new attribute.

network of SM agents, only retaining the symbolic mapping, to investigate the effect of symbolic mapping in vocabulary expansion.

Table 5 and Table 6 show the results of the two experiments. While agents can learn good language in the small environment, they can also achieve good performance in the bigger environment now via vocabulary expansion. This demonstrates that language can evolve to become more complicated as the environment develops, and this process is crucial for agents to learn language in complex environments. The results also reveal that SM agents are better at vocabulary expansion as they can not only express new attributes with the help of new symbols, thus achieving higher success rate, but also use the symbols more compositionally. Note that the reinitialized model performs close to the not reinitialized model, showing that symbolic mapping plays an deterministic role in vocabulary expansion. From this perspective, symbolic mapping is good for language development.

We present an example of the frequencies of different attribute values observed by LSTM and SM agents corresponding to four new symbols in Figure 3. SM agents mainly use the new symbols to express values of the new attribute, showing good compositionality. In contrast, LSTM agents fail to use the new symbols to express accurate meanings after vocabulary expansion.

## 5 CONCLUSION

In this paper, we argue that a process from simplicity to complexity is a natural way to help multi-agent language learning. We have proposed *symbolic mapping* as a basic component of an agent's communication system, and implemented it in LSTM-based agents. This architecture can be applied in different kinds of interactions, so that it can help realize language transfer across different tasks. We conduct experiments about *task transfer* and *vocabulary expansion*, and the results show that learning from simplicity to complexity indeed helps, while symbolic mapping greatly promotes the effect of these two processes. We conclude that symbolic mapping is not only good for language transfer, but also good for language development.

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

## A  TRAINING AND IMPLEMENTATION DETAILS

In all of our experiments, each agent's LSTM has a hidden state of size 50, the dimensions of the hidden layers of all MLPs are the same as their input size, and the entropy regularization parameter $\lambda_H$ is set to 0.05. We train LSTM and SM agents for 10000 epochs in description game and 20000 epochs in discrimination game, unless the agents achieve 100% success rate ahead of time.

The LSTM agents are implemented as LSTM networks with hidden states of size 50. When an LSTM agent observes an object, a linear layer maps the input embedding into the agent's initial hidden state $h_0$. When speaking, we map the agent's hidden state into a probability distribution over the whole vocabulary with an MLP and a softmax function, and we randomly sample a symbol from the probability distribution. The generated symbol will then be fed back into the LSTM. The decision network is the same as SM agents.

We modify the neural iterated learning algorithm to apply iterated learning in our setup. The IL agents' architecture are the same as LSTM agents. The algorithm runs for several generations, and there are three phases in each generation: learning phase, interacting phase and transmitting phase. At the beginning of each generation, all agents are randomly initialized. When training description game, in the learning phase, each agent in the community learns from data collected in the previous generation with cross-entropy, and the shared listener is pre-trained with REINFORCE by interacting with the pre-trained agent community. In the interacting phase, the agent community plays description game with the shared listener and they are trained the same way as LSTM agents. In the transmitting phase, all objects are fed to each speaking agent, and the corresponding messages generated are stored in a dataset for the next generation. When training discrimination game, in the learning phase, two agents are randomly sampled to learn dialogs with supervised learning from data collected in the previous generation, and the rest agent is pre-trained with REINFORCE by interacting with the pre-trained other two agents. In the interacting phase, the agent community plays discrimination game and they are trained the same way as LSTM agents. In the transmitting phase, two agents are randomly sampled, and the whole training set is fed to them to collect the generated dialogs into a dataset for the next generation. In description game training, we set generation number to 20, pre-train iteration number to 2000 for supervised learning and 3000 for reinforcement learning. We train agents for 2000 epochs in the interacting phase. In discrimination game training, we set generation number to 10, pre-train iteration number to 40000 for supervised learning and 100000 for reinforcement learning. We train agents for 4000 epochs in the interacting phase. We tried a set of hyperparameters and use the ones with the best performance.

## B  EXAMPLES OF THE LEARNED SYMBOLIC MAPPING AND COMMUNICATION PROTOCOL

To show what symbolic mapping learns and how it helps task transfer, we conduct the task transfer experiment on a smaller dataset $\mathcal{D}_{2,(3,3)}$ and present here some examples. We refer to the attributes as *color* and *shape*, and each of them has 3 values (*i.e.*, red, green, blue, triangle, square, circle). The vocabulary size is set to 6, the message length is set to 2 in description game and the upper limit for the number of dialog rounds in discrimination game is $T_{\max} = 2$.

Examples of the learned symbolic mapping in the agent community is shown in Table 7 and Table 8. They verify that symbolic mapping is not changed greatly across two tasks, so the learned language can be transferred. In both games, all agents associate symbol '0' with attribute 'green', '1' with 'circle', '2' with 'blue' and 5 with 'square', which presents good compositionality and symmetry.

Table 7: The learned symbolic mapping of the three agents in the community when playing with a shared listener in description game on dataset $\mathcal{D}_{2,(3,3)}$.

| | red | green | blue | | red | green | blue | | red | green | blue |
|---|---|---|---|---|---|---|---|---|---|---|---|
| triangle | 3,4 | 0,3,4 | 2,3,4 | triangle | 3,4 | 0,3,4 | 2,3,4 | triangle | 3,4 | 0,3,4 | 2,3,4 |
| square | 5 | 0,5 | 2,5 | square | 5 | 0,5 | 2,5 | square | 5 | 0,5 | 2,5 |
| circle | 1 | 0,1 | 1,2 | circle | 1,4 | 0,1 | 1,2 | circle | 1 | 0,1 | 1,2 |

Table 8: The learned symbolic mapping of the three agents in the community when playing discrimination game after they have learned to accomplish description game.

| | red | green | blue | | red | green | blue | | red | green | blue |
|---|---|---|---|---|---|---|---|---|---|---|---|
| triangle | 3,4 | 0,3,4 | 2,3,4 | triangle | 3,4 | 0,3,4 | 2,4 | triangle | 3,4 | 0,3,4 | 2,3,4 |
| square | 4,5 | 0,3,4,5 | 2,3,5 | square | 4,5 | 0,5 | 2,5 | square | 3,4,5 | 0,3,5 | 2,3,5 |
| circle | 1,4 | 0,1,4 | 1,2 | circle | 1,4 | 0,1,4 | 1,2 | circle | 1,3,4 | 0,1,3 | 1,2,3 |

Table 9: The learned communication protocols of the three agents in the community when playing with a shared listener in description game on dataset $\mathcal{D}_{2,(3,3)}$.

| | red | green | blue | | red | green | blue | | red | green | blue |
|---|---|---|---|---|---|---|---|---|---|---|---|
| triangle | 3,4 | 0,3 | 2,3 | triangle | 4,4 | 0,4 | 4,2 | triangle | 4,4 | 0,4 | 2,4 |
| square | 5,5 | 5,0 | 5,2 | square | 5,5 | 5,0 | 5,2 | square | 5,5 | 5,0 | 5,2 |
| circle | 1,1 | 1,0 | 1,2 | circle | 1,1 | 1,0 | 1,2 | circle | 1,1 | 1,0 | 1,2 |

Table 10: The learned communication protocols of the three agents in the community when playing discrimination game after they have learned to accomplish description game.

| | red | green | blue | | red | green | blue | | red | green | blue |
|---|---|---|---|---|---|---|---|---|---|---|---|
| triangle | 4 | 0 | 2 | triangle | 3 | 0,4 | 2,4 | triangle | 3 | 0 | 2 |
| square | 4,5 | 0 | 2,5 | square | 4,5 | 0,5 | 2,5 | square | 3,5 | 0,5 | 2,5 |
| circle | 1 | 0 | 1,2 | circle | 1,4 | 0,1 | 1,2 | circle | 1,3 | 0,1 | 1,2 |

Symbol '3' and '4' have relatively ambiguous meanings, which is changed between two tasks, but they mainly cover the attributes 'red' and 'triangle' which cannot be expressed by other symbols. So agents can form compositional structure in symbolic mapping through emergent communication, and the properties like compositionality and symmetry shown in symbolic mapping can explain why symbolic mapping helps language learning through task transfer and why the learned language properties in simple tasks can be maintained in complex tasks by SM agents.

We also present the corresponding communication protocols learned by the agents in the experiment in Table 9 and Table 10. As discrimination game can be terminated at any time, agents may not have chance to express complete information. So in Table 10 we only present all symbols that the agent has spoken in different games after observing a specific object in discrimination game.

Compared with Table 7 and Table 8, the communication protocols make use of the compositional words in symbolic mapping faithfully in both games, so the language is indeed transferred across tasks. Besides, good compositionality and symmetry exhibited in description game are also transferred, which helps success rate in discrimination game.

It may seem odd that the first agent only speaks symbol '0' after observing all green objects in discrimination game. We point out that it results from its game policy: it always expresses 'green' and wait the other agent to communicate about the shape. That may explain why we think speaking policy should not be transferred directly like LSTM agents: policies can be specific to tasks, while only more basic components like symbolic mapping can carry general information about a language.

We should also point out that though the third agent associates symbol '3' with all objects in discrimination game in symbolic mapping, it only speaks it when presented with red objects. This may explain why *refdis* can be higher in protocol compared with mapping.

## C    DETAILED ILLUSTRATION OF SYMBOLIC MAPPING

For clarity, we present here a more detailed illustration of symbolic mapping in Figure 4.

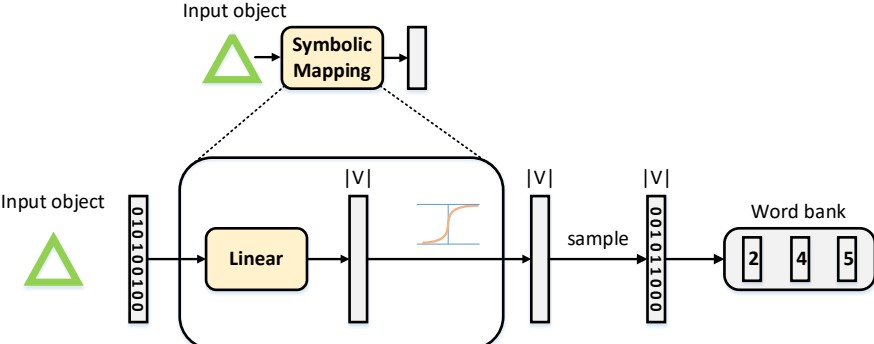

Figure 4: The architecture of symbolic mapping.

