# OpenReview forum: "Multi-Agent Language Learning: Symbolic Mapping"
_ICLR.cc/2022/Conference — ICLR 2022 Submitted_

### Official Review · Reviewer_CC9y · 2021-11-01

**Correctness:** 3
**Technical Novelty And Significance:** 3
**Empirical Novelty And Significance:** 2
**Recommendation:** 6
**Confidence:** 4

**Main Review:**

## Post-rebuttal Edit
In light of the added clarity regarding symbolic mapping, I am upgrading my review from 5 to 6 and correctness from 1 to 3.

## Strengths
- It presents an intuitive yet novel approach to emergent language: transfer from a simpler to more complex task.
- The work is complete in itself but clearly points the way towards future directions.
- The structure of experiments provides the right kind of empirical support for the premise.

## Weaknesses
- The explanation of the core method, symbolic mapping, is unclear, which makes a difficult to evaluate exactly what is happening with the task transfer.
- The experiments use a small number of trials (3) in very small environments which raises concerns about the robustness of the observed effects.
- It seems as if refdis is selected as the metric for compositionality and then discarded in Section 4.2.2 Paragraph 3 with the claim that the communication protocol is compositional anyway despite the low refdis.

## Recommendation
I recommend "reject" in the current form of the paper because even with careful reading, I am not able to clearly understand what symbolic mapping is, a key component of the paper.
I would be happy to change my recommendation with revisions which clarify the explanation of symbolic mapping.
Less important, but running a greater number of trials and giving a rigorous qualitative explanation when compositionality does not line up with refdis would strengthen the paper.

## Justification
The paper up to Section 3.2 is well written and motivated, but Sections 3.2 and 3.3 (and Figure 2) are critical yet unclear.
An understanding of symbolic mapping bridges the conceptual gap from task transfer as a motivated method of doing emergent language to concrete instantiation of the agents in neural networks.
As a result, I cannot determine if the experimental results are trivial or informative.
Direct references to the location of issues in the paper are given in `Additional Comments`.

Specific criteria:
- Correctness: 1
    - Although I cannot judge the experimental design fully until I understand symbolic mapping, I believe that the experiments largely support the claims (i.e., 3).
    - A larger number of trials and a clearer analysis of refdis (in the context of the experiments -- it's formulation is clear) would be needed in addition for a 4.
- Technical Novelty and Significance: 3
- Empirical Novelty and Significance: 2
    - As mentioned above, a larger number of trials is needed.
    - Additionally, the empirical significance is lessened by the fact that only the only environments that work are of the scale $(3,3,3)$ and $(4,4)$.

## Questions
- What exactly is symbolic mapping? Equations and examples can always make things clearer.
- In what way is refdis an inadequate metric (if at all)? Is there a better alternative that could be used for this paper?

## Additional Comments
- `s1`: Contributions should explicitly stated in the introduction
- `s1`: "curriculum learning" should at least be mentioned somewhere since it is a common concept in machine learning
- `s3.1 discrimination game p1`: "is comprised of" -> "comprises"
- `s3.1 discrimination game p2`: use $m_a$ or $m^{(a)}$ instead of $m^a$ -- it took me a minute to realize it was indexing and not an exponent
- `s3.1 description game p1`: Is there any reason the simpler game is second? It would seem to make a bit more sense going first.
- `Figure 2`: This figure is very unclear insofar as the most important details are left unspecified.
    - Do the triangle and the symbolic mapping box have any relation? Or do they just happen to be adjacent?
    - How is the green triangle represented numerically, since it is presumably being concatenated with a vector?
    - What is the symbolic mapping box? What is coming out of it?
    - What are the inputs and outputs to the word bank?
    - Is the word bank generating multiple $s$'s? Why is that?
    - Why are there three copies of the speaking network and why do they appear to feed into the same softmax?
    - What is the purpose of the decision network and what $v_t$?
    - Is the robot face in the top right corner supposed to represent anything?
    - What is the rec network?
    - What is $m$ in the listening model?
- `s3.2 p1`: What does it mean to "sample symbols according to the probability given by the output of the sigmoid function"?
    Please be clear with what the symbols and the word bank are because they are critical to understanding symbolic mapping.
    What are the values in the vector and why do they correspond to "relevance"?
    What is the probability distribution being used here, is it a Bernoulli distribution for each element of the vector or a categorical distribution over the whole vector?
- `s3.2 p2`: Initializing with Gaussian noise seems a bit more typical and could maybe aid in robustness slightly. Just speculation on my part, though.
- `s3.2 p2`: "id" -> "identity"
- `s3.2 p3`: What does it mean for a symbol to be encoded into a one-hot embedding?
- `s3.2 p3`: What is the distribution $pi^i_\text{sp}$?
- `s3.2 p3`: What is the action $v_{t+1}$?
- `s3.2 p3`: "an one-hot" -> "a one-hot"
- `s3.2 p4`: "In description game" -> "In the description game"
- `s3.2 p4`: What does it mean for the speaker to be represented by the "bag-of-words model"?
    I am familiar with BoW in the context of text features, but I am not sure how it is applied here.
    Does that just mean that the listener receives a sum of vectors which the speaker has generate?
- `s3.2 p5`: 3 independent runs seems very small. On the order of 10s seems like a minimum while 100s is even better.
- `s3.4 Compositionality`: I see how positional disentanglement is inappropriate for this task, but BoW disentanglement from the same paper might work just fine -- it is very similar to referential disentanglement as given here. Furthermore, 15 minutes of notebook scribbles leads me to think that it might not suffer from the issue of redundant symbols for a given attribute (i.e., bosdis is still high unlike refdis).
- `s4.1 p1`: "unfixed" -> "not fixed"
- `s4.2.1 p2`: If the experiments are using a shared listener in the dialog setting, wouldn't this undermine the concept of refdis measuring _symmetry_? It seems like it would be measuring something more like consensus or inter-speaker agreement since the agents are not properly symmetric (i.e., independent speaking and hearing models).
- `Table 2`: What is the meaning of "protocol" and "mapping"?
- `s4.2.2 p3`: It seems like the point is being made that compositionality is still present despite the low refdis.
    This undermines the usefulness of the metric.
    If the claim is that the metric does not capture compositionality in this case, a well-illustrated qualitative argument should be made to this end.


**Summary Of The Paper:**

This paper models a new aspect of emergent language: developing language to complete a simple task before transferring the emergent language to a new, more complex task.
It does this by creating curriculum of a simple referential game followed by a slightly harder dialog task.
A traditional architecture does not perform well on this task transfer so the paper introduces _symbolic mapping_ in order to encourage a compositional language which generalizes well.

The following empirical evaluations are performed:
- discrimination (dialog) game only
- description (referential) game then discrimination game with and without symbolic mapping
- expanding vocabulary by transferring to a new task with new attributes to describe

The contributions of the paper are:
- Introducing experiments which look at the progression from a simpler game to a more complex one
- Developing a novel architectural component called _symbolic mapping_ which aids in both transferring to new tasks and in developing a compositional communication protocol
- Introducing a metric referential disentanglement based off of positional disentanglement


**Summary Of The Review:**

The paper presents a novel and motivated setting of task transfer and has the right start for experiments.
On the other hand, the explanation of symbolic mapping, central to linking the setting and experiments, is very unclear.
I intend to change my "reject" to an "accept" if the symbolic mapping is sufficiently clarified.

---

> ### Author Response · Authors · 2021-11-18
> **Response to Reviewer CC9y (1/2)**
>
> Thanks for your comments. We updated the paper accordingly to clarify the symbolic mapping. For details please refer to Section 3.2 and Appendix in the revision. We hope the modifications and the following replies can clarify your concerns.
>
> > What exactly is symbolic mapping?
>
> Symbolic mapping maps an input object to a vector with dimension $|V|$, where each value in the vector corresponds to the degree of relevance between a symbol and the object. Each value is an output of a sigmoid function, so it is also seen as the probability that the corresponding symbol is associated with the object. Symbols are sampled according to the probability using the Bernoulli distribution for each element of the vector and stored as the word bank. Then, the speaking policy selects the symbol from the work bank for communication. The number of symbols in the word bank is not predefined or limited, and during training each symbol has chance to be sampled. So the agents learn the symbolic mapping at the same time when they learn to communicate.
>
> We added simple examples of the learned symbolic mapping in Appendix, showing that after language learning agents can learn to associate observed objects with only a few symbols, each of them refering to the attribute value of the object. We also added a figure to illustrate the symbolic mapping more clearly in Appendix in the revision.
>
> Please let us know whether we have made it clear to you or if you have addtional questions about symbolic mapping.
>
>
>
> > In what way is refdis an inadequate metric (if at all)? Is there a better alternative that could be used for this paper?
>
> Refdis works when we only consider disentanglement for compositionality, so if more complex forms of compositionality are also considered, refdis may be inadequate. Most other metrics are designed according to specific focuses on compositionality, maybe different from ours and not suitable in our paper. Some metrics can be hard to calculate because of different setups. There may be other metrics evaluating the same property, but we don't think they will be better than refdis, since refdis is adequate in our setting.
>
>
>
> > Table 2: What is the meaning of "protocol" and "mapping"?
>
> > It seems as if refdis is selected as the metric for compositionality and then discarded in Section 4.2.2 Paragraph 3 with the claim that the communication protocol is compositional anyway despite the low refdis.
>
> Metrics are calculated on both communication protocol and symbolic mapping. It is a misunderstanding here. For SM agents in the task transfer experiment, the refdis is high calculated on their communication protocols, but relatively low (actually still higher than the LSTM baseline) calculated on the symbolic mapping. So this part means the symbolic mapping is not a highly compositional representation here because of the redundant symbols.
>
>
>
> > The experiments use a small number of trials (3) in very small environments which raises concerns about the robustness of the observed effects.
>
> We are running the experiments for more trials, and will update the paper once obtain the results.

---

> > ### Comment · Reviewer_CC9y · 2021-11-19
> > **Understanding symbolic mapping**
> >
> > Thank you for the reply.
> > Symbolic mapping is becoming gradually clearer.
> >
> > So an input object in $\mathcal D_{(3,2)}$ would be represented as a vector of length $3+2=5$?
> > Would it them be able to take on the values:
> > ```
> > [0 0 1 0 1]
> > [0 1 0 0 1]
> > [1 0 0 0 1]
> > [0 0 1 1 0]
> > [0 1 0 1 0]
> > [1 0 0 1 0]
> > ```
> > ?
> >
> > And let's say that post-sigmoid is sampled to yield `[0 1 0 1]` (from 4 independent Bernoulli sample);
> >     does that mean that the word bank will now be `[[0 1 0 0], [0 0 0 1]]`?
> >
> > Finally, how do you backpropagate through the sampling?
> > Do you use straight-through estimation (i.e., just treat the parameter of the Bernoulli distribution as the value you are backpropping through)?

---

> > > ### Author Response · Authors · 2021-11-20
> > > **Response to the questions**
> > >
> > > > So an input object in $\mathcal{D}_{(3,2)}$ would be represented as a vector of length 3+2=5? Would it them be able to take on the values...
> > >
> > > Yes. (But the notation should be $\mathcal{D}_{(2,(3,2))}$)
> > >
> > > > And let's say that post-sigmoid is sampled to yield `[0 1 0 1]` (from 4 independent Bernoulli sample); does that mean that the word bank will now be `[[0 1 0 0], [0 0 0 1]]`?
> > >
> > > Yes.
> > >
> > > > Finally, how do you backpropagate through the sampling...
> > >
> > > We do not backpropagate through the sampling. Instead, the output of the symbolic mapping is treated as part of the agent's actions and we train it using REINFORCE.

---

> ### Author Response · Authors · 2021-11-18
> **Response to Reviewer CC9y (2/2)**
>
>
>
> > Is there any reason the simpler game is second?
>
> Our main target is to conduct language learning in the difficult game, and the simple game is introduced to perform task transfer which helps language learning in the difficult one. This order also matches the order in the experiments.
>
> > Figure 2
>
> We modified the figure to make it clear. Thanks for these comments.
>
> > > Do the triangle and the symbolic mapping box have any relation?
>
> The triangle is the input object here, namely the input of the symbolic mapping.
>
> > > What are the inputs and outputs to the word bank?
>
> Word bank is not a neural network. It consists of the sampled symbols from the symbolic mapping. You can see it as the output of the symbolic mapping.
>
> > > Why are there three copies of the speaking network and why do they appear to feed into the same softmax?
>
> Actually there is only one speaking network. We depict several copies since each symbol in the word bank is concatenated to the hidden state and fed into the speaking network. So each symbol has an output given by the shared speaking network, and all these outputs are fed into the softmax.
>
> > > What is the purpose of the decision network and what vt?
>
> The decision network receives the hidden state and the object, and produces and action $v_t$. The agents have $n+2$ actions here, where the first corresponds to continuing the dialog while the left $n+1$ ends the dialog and gives out the answer to this game (the predicted label $l_p$ ).
>
> > > Is the robot face in the top right corner supposed to represent anything?
>
> It just means the agent, implying that the depicted models belongs to a single agent.
>
> > > What is the rec network?
>
> It is the reconstruction networks used in the description game.
>
> > > What is m in the listening model?
>
> The received message sent by the speaker in description game.
>
> > What does it mean to "sample symbols according to the probability given by the output of the sigmoid function"?
>
> Some has been explained before. The output of the linear layer of symbolic mapping is activated by the sigmoid function, so the values in the vector are between 0 and 1. Each value corresponds to a symbol, and higher value means higher relevence between the symbol and the input object. The probability distribution used here is a Bernoulli distribution for each element of the vector.
>
> > What does it mean for a symbol to be encoded into a one-hot embedding?
>
> One-hot embedding is easier for learning in practice.
>
> > What is the distribution $\pi_{sp}^i$
>
> It is the distribution given by the softmax function, whose input are the outputs of the shared speaking network.
>
> > What does it mean for the speaker to be represented by the "bag-of-words model"?
>
> We represent the message sent by the speaker using BoW, so the resulting vector contains the counts of symbols in the message.
>
> > BoW disentanglement from the same paper might work just fine
>
> Bosdis considers information given by the symbol counts, which we think is not appropriate in our setting. In our dialogs messages are not sent completely at a time, but one symbol each turn, so it is unlikely and also not expected that the counts of symbols are informative.
>
> > If the experiments are using a shared listener in the dialog setting, wouldn't this undermine the concept of refdis measuring *symmetry*?
>
> By symmetry we mean the language protocols are symmetric between different agents, similar to consensus as you mentioned.
>
> > What is the meaning of "protocol" and "mapping"?
>
> Metrics are calculated on both communication protocol and symbolic mapping.
>
> > It seems like the point is being made that compositionality is still present despite the low refdis.
>
> It is a misunderstanding here. For SM agents in the task transfer experiment, the refdis is high calculated on their communication protocols, but relatively low (actually still higher than the LSTM baseline) calculated on the symbolic mapping. So this part means the symbolic mapping is not a highly compositional representation here because of the redundant symbols.

---

### Official Review · Reviewer_Litw · 2021-11-01

**Correctness:** 2
**Technical Novelty And Significance:** 2
**Empirical Novelty And Significance:** 3
**Recommendation:** 6
**Confidence:** 4

**Main Review:**

**Weaknesses:**
1. Unclear that SM, refdis/refdiv, and the reinforcement learning tasks are sufficiently original to stand alone.
2. Missing significant literature.
3. General issues with clarity

**Strengths:**
1. Well-motivated and well-designed experiments.
2. Results make a strong case for further work in intermediate symbolic representations and support prior research!

**Recommendation:**
Based on the above reasons, especially due to lack of citations/originality, I vote to reject the paper in its current form. However, since there are strong empirical contributions, if the authors revise the paper to include the relevant prior work (details below), I'd willing to change my vote to accept the paper.

**Questions and Additional Comments:**
(organized by originality, technical quality, and clarity)

*Originality============================================================*

In general, the paper can benefit from a more thorough discussion on how the new research differs from past work. There are several major areas in the paper where the previous literature is inadequately cited (see points 1-4). It's unclear that the authors' modelling contribution and reinforcement learning tasks are sufficiently original to stand alone. However, the experiments showing that SM is beneficial in task transfer are important in that they support prior findings, and the experiments on vocabulary expansion are novel, to my knowledge.

(1) The main idea of symbolic mapping, which is mapping raw input to an intermediate symbolic representation, is not new and should be discussed in relevant work. See work such as "Towards Deep Symbolic Reinforcement Learning" (Garnelo, et al 2016) and "Reconciling deep learning with symbolic artificial intelligence: representing objects and relations" (Garnelo, et al 2019) for a starting point.

(2) The idea of bootstrapping in emergent communication from simpler tasks to harder tasks is not new and should be properly cited in section 2 (I suggest removing the statement "orthogonal to existing work"). See "Developmentally motivated emergence of compositional communication via template transfer" by Korbak, et al (2019) and its references for a starting point.

(3) The literature in Section 1 on language evolving from simple to complex tasks can be greatly expanded. See Deacon's "The Symbolic Species" and other references in Korbak, et al (2019) for a starting point.

(4) The concept of refdis is very similar to context independence introduced in "Emergence of Communication in an Interactive World with Consistent Speakers" by Borgin et al (2018) and discussed in "On the Pitfalls of Measuring Emergent Communication" by Lowe et al (2019). It would be good to differentiate context independence from refdis, discuss with a citation in section 3.4.

Nit:
(5) Rewording the explanation of Eq. (1) to be less similar to the explanation of Eq. (1) in Chaabouni, et al (2020).

*Technical details ============================================================*

Overall, the authors explain technical details and modelling choices thoroughly, and well-motivate the training scheme/experiments. In particular, the LSTM ablation results are strong in that they suggest SM produces a good intermediate object representation. While the big-picture is technically sound, I disagree with some of the claims in the paper and would like to see a deeper discussion of referential disentanglement. I recommend editing the verbiage to be more task-specific instead of general (see point 1).

(1) Section 3.3: "symbolic association which is nearly orthogonal to tasks…" I disagree. For example, in grounded language games, symbolic association is closely linked to the task. It is important to specify that this claim applies to *this specific setting* rather than in general tasks as implied.

(2) The discussion of compositionality in Section 3.4 needs further development. Why does referential disentanglement, which is similar to context independence, capture compositionality in this setting? What are the limitations of referential disentanglement? Note that referential disentanglement works in this setting because the semantic composition rule of the attributes is intersective and not order-dependent.

(3) Section 4.1: "agent cannot predict… when the dialog will be terminated." Is this problem solved when fixing the number of turns instead of letting agents terminate the game?

(4) Section 4.3: was NIL tested in vocab expansion?

*Clarity ============================================================*

The implementation of the model and experiments are mostly clear. However, there are some statements that I found vague, some claims that can be better argued, and some adjustments that can improve the flow of the paper.

(1) In Section 3.1, it's unclear what the turn taking structure is until Section 3.2. It would help to include a line saying players explicitly take turns unless the game is terminated.

(2) The first two lines on pg. 4 are hard for me to understand. Why does detecting the difference between objects imply the agents may not be clear what their partner's symbols mean? I would think that it's never guaranteed that the agent understands the meaning of their partner's symbols.

(3) In Section 3.1, Description Game, reconstruction model is used first and defined later. I would include a line explaining what a reconstruction model is.

(4) Section 3.3: "though the mapping is simple… help maintain language properties across tasks." What is intended by "language properties"?

(5) Section 3.3: in the explanation of vocabulary expansion, the word "vocabulary" is overloaded. It's not clear upon initial reading that vocabulary expansion is increasing the size of the word bank rather than the base vocabulary.

Nit:
(6) We fix the message length to n (unclear upon first reading why it's n) -> We fix the message length to n, corresponding to one symbol per attribute.

(7) Please state both bounds for both refdis and refdiv.

(8) Table 1: second and third column -> first and second column

(9) Section 4.3: "We have empirically proved that language can evolve from simple tasks to difficult tasks" is too general. Change to "We have shown that our agents' language/communication protocol evolved in task transfer,.." or something more specific.

(10) Figure 3: it's hard to tell blue/green and red/orange apart.

(Not important for this review, but please clean up grammatical mistakes for future iterations.)


**Summary Of The Paper:**

This paper has several main contributions: (1) A novel architecture, symbolic mapping (SM), (2) Showing SM outperforms vanilla LSTM and NIL in terms of success rate, degree of compositionality, and degree of symmetry on the description game and discrimination game after bootstrapping,(3) Showing SM aids vocabulary expansion moving from a smaller to a larger vocabulary and attribute space.

SM, a linear module that maps an object to a subset of the symbol vocabulary, is intuitively an agent's internal object representation or "word bank". A recurrent speaking module then selects the speaker agent's utterance from their word bank. SM associates object to symbols in a task and speaker-agnostic way. This not only facilitates the transfer of these object-symbol mappings to different tasks but also expanded vocabularies.

Theories of language emergence posit that human communication evolved to be more complex by adapting to increasingly complex settings. Following this hypothesis, the authors define two emergent communication games, a description game which is easier and discrimination game which is harder, and use SM to facilitate the transition between the two. They quantify performance using training and test success rate, as well as refdis (compositionality) and refdiv (symmetry). They find that SM outperforms a vanilla LSTM and NIL in the description game in all metrics. They show that all three models perform poorly trained from scratch in the discrimination game, then that their performance improves after pre-training speaker agents on the description game. Notably, when bootstrapping from the description game, SM achieves a higher success rate while also exhibiting a higher degree of compositionality and symmetry than the other two models. This supports the effectiveness of SM as an internal object representation for the agent.

Finally, the authors conduct a vocabulary expansion experiment. They first show that both an LSTM and SM agent learn poorly  from scratch on a large attribute space and vocabulary size. Then, they show that all agents perform well on the same task after bootstrapping from a smaller attribute space and vocabulary size. In particular, SM (whether or not reinitialized) outperforms LSTM after transfer.

**Summary Of The Review:**

The current version of the paper greatly overstates the novelty of its technical contributions (SM, refdis, task transfer). To me the primary contribution is therefore the synthesis of prior technical contributions, the strong experimental result, and the case for further research into symbolic representations. Only on the condition that the authors substantially revise their discussions of SM, refdis, and task transfer to credit prior work, I would recommend the paper for acceptance.

---

> ### Author Response · Authors · 2021-11-18
> **Response to Reviewer Litw**
>
> Thanks for your comments. According to your suggestions in the review, we have updated the paper to substantially revise the disccusions of SM, refdis and task transfer to credit prior work. For details please refer to the related work and metrics part in the revision. In the following, we provide the itemized responses to your comments.
>
> > Originality
>
> These suggestions are really helpful, and we have updated the paper accordingly.
>
> > > symbolic representation
>
> Symbolic mapping can be seen as a kind of symbolic representation in its function. Different from prior work, symbolic mapping is learned and constructed through emergent communication instead of representation learning techniques and is trained end-to-end by RL. That means agents form the symbolic representation when learning to communicate.
>
> > > from simpler tasks to harder tasks
>
> Different from template transfer, where a hard task is split into several parts and the transferred agent is the listener, we explore language transfer from simple interactions to different tasks involving more complex communication forms, and the speaker is not reinitialized so that the language evolution is consistent.
>
> > The literature in Section 1 on language evolving from simple to complex tasks can be greatly expanded.
>
> Thanks for the suggestion. We will expand this part after sufficient corresponding literature reading.
>
> > > refdis and context independence
>
> Refdis is modified from posdis and removes the positional information. The key idea is similar between refdis and context independence, and the difference is that refdis focuses on symbols and is normalized by  their occurrence frequency while context independence focues on the alignment between symbols and concepts.
>
> > Technical details
>
> > > It is important to specify that this claim applies to *this specific setting* rather than in general tasks as implied
>
> It's a good suggestion and we updated the descriptions.
>
> > > The discussion of compositionality in Section 3.4 needs further development.
>
> As you mentioned, refdis works in our setting since we only consider disentanglement for compositionality, so if each symbol refers to a specific attribute accurately it achieves high refdis. If more complex forms of compositionality are also considered, refdis may not work well.
>
> > > Is this problem solved when fixing the number of turns instead of letting agents terminate the game?
>
> We use unfixed number of turns to create a more natural dialog setting, and we hope our method can help address the introduced difficulty.
>
> > > was NIL tested in vocab expansion?
>
> No. NIL is a training framework for compositionality, and the agent architecture is the same as LSTM agents. Since we try to verify the beniefit of the SM architecture in vocabulary expansion, we think the LSTM baseline is enough for the experiment.
>
> > Clarity
>
> Thank you for the thoughful suggestions and we have adjusted the paper.

---

> ### Author Response · Authors · 2021-11-24
> **Follow-up**
>
> As the discussion will end soon, we would like to know whether our responses have addressed your concerns. Please let us know if you have further questions.

---

> > ### Comment · Reviewer_Litw · 2021-11-29
> > **Updating score to 6**
> >
> > Thank you for your responses. I would take another look to edit grammar, however (noticed a couple of mistakes remaining after skim).

---

> > > ### Author Response · Authors · 2021-11-30
> > > **Response**
> > >
> > > Thanks for your careful review, and we will address grammatical mistakes.

---

### Official Review · Reviewer_Z2BE · 2021-11-03

**Correctness:** 3
**Technical Novelty And Significance:** 4
**Empirical Novelty And Significance:** 3
**Recommendation:** 6
**Confidence:** 4

**Main Review:**

The paper is well motivated and is easy to understand. The authors perform an extensive search over the relevant metrics used for computing compositionality and systematic generalization in an emergent communication setting. Although, the authors show promising results on the small 'toyish' dataset, there is no evidence or discussion on whether the findings would scale to more complex domains. The two different games used in the paper still only use symbolic data with only a limited number of training samples.

The task transfer experiment is not clearly stated. The authors state that symbolic mapping provides knowledge about the learned language implicitly. It would be better to demonstrate this with a small example on what exactly transfers from learning the simple solution to solving a more complex task. Moreover, the authors refer to increasing the complexity of the task to increasing the number of types of an attribute. How would this approach scale if we add a new dimension of complexity? Would it require fewer game interactions or samples than training an agent from scratch? More empirical evidence is needed to support the claim that the proposed module would scale to any complex task, or state how such a complex task should be designed.

The authors state that training agents in the discrimination game is harder than expected and the effect of symbolic mapping is not really pronounced in this game. A qualitative example would be helpful to highlight the possible issues with this game as compared to the description game.



**Summary Of The Paper:**

The paper talks about introducing a new learning paradigm to train compositional and symmetric emergent languages while the tasks increase in complexity as the training progresses. The main motivation comes from works in psychology that show that human languages are also evolved from a simpler task to gradually increasing the task complexity. The proposed symbolic mapping module is shown to perform better than training agents directly on the complex task, thereby showing the importance of using past strategies to learn a solution for the new complex task.

**Summary Of The Review:**

The authors propose a novel module that helps in training better compositional and symmetric language using a curriculum from simple to complex tasks. The paper is well-written and the experiments show that the proposed method indeed supports the hypothesis although the experiments are only performed on a simple environment. The complexity of the tasks can be varied along different axis but only a few options are explored to extend this.

---

> ### Author Response · Authors · 2021-11-18
> **Response to Reviewer Z2BE**
>
> Thanks for your comments and suggestions.
>
> > The task transfer experiment is not clearly stated...
>
> It is a good idea to present some examples to show what the symbolic mapping learns in the tasks and how it helps language learning in the complex game. We updated the appendix to include some examples. Scaling to new dimensions of complexity and other complex tasks are interesting points and we will make such attempts in the future.

---

### Official Review · Reviewer_ZcF5 · 2021-11-04

**Correctness:** 4
**Technical Novelty And Significance:** 2
**Empirical Novelty And Significance:** Not applicable
**Recommendation:** 5
**Confidence:** 3

**Main Review:**

Papers like this are hard for me to evaluate. There has been quite a bit of previous work on looking at what environmental pressures and inductive biases lead to the emergence of compositional communication. In these papers, the environments considered are so simple that the systems developed are unlikely to be practical for any near-term applications. The contributions in these papers are instead scientific -- do the experiments help us understand something new about how simulated languages develop? Are the experiments rigorously performed?

I would say this is not a bad paper. The paper uses an environment that, while simple, is in fact a bit more complicated than environments in previous emergent communication papers. I would lump their 'task transfer' and 'vocabulary expansion' results under the general banner of curriculum learning -- if you first train on a simpler task, then train on the more complicated task, you will do better at the more complicated task. This is pretty intuitive, I'm a bit surprised that no previous paper to my knowledge has made this specific point (though it may exist), but I think this is a positive contribution of the paper.

The paper also proposes a fairly specific LSTM-based architecture (SM) that further improves performance. I don't find this result particularly compelling -- given that it is mostly an engineering contribution, I'd want to see it tried on a broader range of tasks that we care about. One of the claimed benefits of SM from section 3.3 is that the architecture allows us to do 'vocabulary expansion', but the results indicate that vocabulary expansion seems to help the base LSTM policy just as much as the SM policy. I don't find the remaining arguments and intuitions in 3.3 general or compelling, but I may be missing something.

Ultimately, if we take the main contribution of the paper to be 'forms of curriculum learning help compositional languages to emerge', I don't think this paper presents experiments that are quite extensive enough for the reader to really understand *how* curriculum learning helps. I'd want to see more ablations, e.g. looking at multi-step curricula (the paper only considers 2 steps), effect of initial task difficulty, etc..

Overall, I think the paper is not quite at the level of acceptance, and would currently recommend rejection.

**Summary Of The Paper:**

This paper investigates methods for emergent communication. Specifically, it looks at inductive biases that can contribute to the emergence of compositional and symmetric languages. The main environment the paper studies is a multi-turn dialog game, where two agents are shown objects that differ by 0 or 1 attribute and have to agree on the differing attribute.

The paper makes 2 main contributions:

1) It investigates the role of curriculum learning, and shows that first training on a simpler task -- a referential description game (task transfer), or the discrimination game with fewer properties (vocabulary expansion) -- helps agents learn compositonal languages on the broader discrimination game.

2) It proposes a new agent architecture, 'symbolic mapping', which can help agent performance and increase compositionality.

**Summary Of The Review:**

While the paper shows that curriculum learning can help compositional language emergence, which is interesting, I'd want to see more extensive experiments and ablations to recommend acceptance.

---

> ### Author Response · Authors · 2021-11-18
> **Response to Reviewer ZcF5**
>
> Thanks for your comments.
>
> We'd like to argue that our approach is *not* simply to use curriculum learning in emergent communication. While curriculum learning focuses on designing an organized training procedure to guide agents learn hard tasks more quickly and easily, our approach focuses on providing agents with an easy start to form a communication protocol, and investigating whether language can be transferred or evolve to more complicated environments. Curriculum can bring performance improvement but we also care about whether language properties can be maintained. The idea that training on a simpler task can ease the learning in more complicated tasks is indeed similar to curriculum learning and intuitive, but it is *not* trivial in emergent communication because language transfer across different tasks is *understudied*, and our LSTM baseline shows that a straightforward implementation performs not well.  Therefore, we think the experiments in our paper can already help us achieve our main goal.
>
> As for the choice of the initial task, inspired by psychological studies, we hypothesize that language should be first originated from simple interactions like pointing and pantomiming, so we refer to games similar to these interactions as 'simple tasks', such as simple referential games, and we refer to games involving more complex communication forms as 'complex tasks', such as dialog games. Besides, inspired by the fact that vocabulary in natural language changes continually, we investigate task with fewer attributes to task with more attributes as another dimension about 'simple to complex'. As shown in Table 6 and Figure 3, SM agents achieve higher success rate (success rate 100% vs. 84%) and more compositional communication (*refdis* 0.91 vs. 0.47), and use the new symbols to express values of the new attribute. These empirically support the claim about symbolic mapping in Section 3.3.
>
> There can be other choices or more detailed learning routes of 'from simple to complex', but more cognitive science findings should be involved, so we tend to leave this for future work.

---

### Official Review · Reviewer_rYPh · 2021-11-07

**Correctness:** 2
**Technical Novelty And Significance:** 1
**Empirical Novelty And Significance:** 1
**Recommendation:** 3
**Confidence:** 2

**Main Review:**


I have a number of larger issues with the paper. Some of this may stem from me not being able to understand the paper. The writing isn't very clear. Concepts are seemingly introduced in a random order. The authors move around between architectures, tasks, training regimes - which made it very hard to follow for me.

The paper is trying to make too many different points including a "new" architecture, a "new" game, a "new" curriculum approach. Some of these seem trivial or known in prior work.

It's quite known in RL more generally and in language emergence more broadly that curriculums are helpful. I don't think the paper really adds anything interesting to that discussion.

Unfortunately, I did not understand well how the agent architecture works. It's not clear how the word bank works and what the speaking networks and reconstruction networks are. But from what I understand - the symbolic mapping is basically an MLP that looks up words for objects. It's these words then that are fed to the recurrent sentence generator. This seems like a prior that works in the particular setting here. However, it also seems like a trivial extension.


Important baselins such as IL are not explained in the paper.

Some major details are missing. E.g. the different MLPs such as the reconstruction network - how many layers etc

Other Comments and Questions

was originated -> originated

The definitions around objects in 3.1 don't seem to make sense. If a \in {1,2,...,n} and there are n attribues in each object - are you saying that all objects are the same? What's m? It seems to be the number of values per attribute. but you already said that the value of a is one of these {1,2,...,n}. That's quite confusing
Are you saying that all attributes have the same number of possible values in your experiments?

It's not clear to me what is the word bank and what does it consist of.

What is a reconstrution network? The term is used once and never reappears. Is it the rec networks in your figure? Are they per attribute?

What aren the dimensions for your different MLPs

Why do you use $s$ and $w_i$ for the word?

"then both agents success" -> "then both agents succeed"?

I think it's not useful to use the term discrimination game and then suggest that it's more difficult since it requires dialogue. Discrimination games have been introduce in the 90s and they are referential and they do not necessarily require dialogue

Similarly people have used the term "decscription game" in emergent communciation research since the 90s and it is precisely not a referential game in the classical sense.

**Summary Of The Paper:**

The paper proposes symbolic mapping. The main point of this method is to associate objects with words in processing before making pragmatic decisions about what should be said. THis is then part of a larger recurrent architecture that produces words to describe or discrimate objects. The architecture is trained using REINFORCE - in different scenarios. Another point the paper is trying to make is that training the symbolic mapping on a simple scenario and the explanding complexity helps. The method is evaluated for success but also in terms of its ability to create compositional and symmetric language.

**Summary Of The Review:**


The paper unfortunately suffers from some unclear writing as well as some lengthy unimportant descriptions and some important aspects that are missing

PRO
* interesting domain with lots of problems
* main claim of language emergence from simple to complex is illustrated by one example. But this is such a general claim - it would require a lot more experiments
* the main claim of the paper is not really a technical claim

CONS
* some key explanations are unclear.
* many less important explanations are lengthy and convoluted making it difficult to follow
* somewhat trivial domain (that is pretty common though in this line of research)

---

> ### Author Response · Authors · 2021-11-18
> **Response to Reviewer rYPh**
>
> It is unfortunate that you are not enable to understand the paper well. According to your comments, we have revised the manuscript. We think the current version is much clearer than previous one on many aspects. Please let us know if you have additional comments. In the following, we provide the itemized responses to your comments.
>
>
> > It's quite known in RL more generally and in language emergence more broadly that curriculums are helpful...
>
> We argue that though it is known that curriculum learning is helpful, it does not mean that task transfer and vocabulary expansion are trivial in emergent communication. As mentioned in the paper,  language is different from specific speaking policy since it is an abstract capability, so rather than transferring speaking policies, we investigate whether language can be transferred by neural network agents across different tasks, and how to keep language properties through the transfer, which is *understudied*. Therefore we design symbolic mapping. Empirical results for SM agents and LSTM agents demonstrate this point -- curriculum can indeed bring the performance improvement, but **vanilla LSTM cannot well maintain the good language properties learned in the simple task, while symbolic mapping helps to accomplish difficult tasks and maintain good language properties.**
>
>
> > Unfortunately, I did not understand well how the agent architecture works. It's not clear...
>
> Symbolic mapping is a neural network that associates symbols with the input object. Word bank consists of symbols sampled from the association. Then symbols in word bank are selected by the speaking policy for communication. We need to argue that symbolic mapping is not a trivial extension, but an intermediate symbolic representation, which is also investigated in several previous studies (see revision for details) for other deep learning tasks. Unlike these studies, we investigate symbolic mapping from the perspective of language evolving from simple to difficult tasks. Moreover, we also added a figure to illustrate symbolic mapping in Appendix in the revison.
>
>
> > Important baselins such as IL are not explained in the paper.
>
> You can find the explanation and implementation details in Appendix.
>
>
> > Some major details are missing. E.g. the different MLPs such as the reconstruction network - how many layers etc.
>
> The MLPs in our architecture have 2 layers. We have updated the manuscript to address these.
>
>
> > The definitions around objects in 3.1 don't seem to make sense...
>
> $a$ is the index of the attribute and $m^a$ is the number of values of attribute $a$. For example, when $a$ corresponds to attribute 'color', then $m^a=3$ means there are 3 diffrent colors in the dataset. We have change it to  $m^{(a)}$ to avoid the confusion. Please see game settings for details in the revision.
>
>
> > It's not clear to me what is the word bank and what does it consist of
>
> Word bank consists of the symbols sampled from the symbolic mapping. It is not a neural network.
>
>
> > What is a reconstrution network.....
>
> Yes, they are the rec networks in Figure 2, and they are linear layers, which is mentioned in the paper. They are per attribute. We have revised the manuscript to make it clear.
>
>
> > What are the dimensions for your different MLPs
>
> Please refer to Appendix for details.
>
>
> > Why do you use s and wi for the word?
>
> $w_i$ denotes the word bank.

---

### Decision · Program_Chairs · 2022-01-20

**Decision:**

Reject

**Comment:**

This manuscript presents a novel approach to learning a shared language between multiple agents.

In general, reviewers had difficulty understanding the symbolic mapping component. For such a critical part of the manuscript, questions by multiple reviewers were extremely basic, asking what symbolic mapping even is. Authors did clarify this in the discussion and updated the manuscript, but further improvements to the manuscript are warranted.

Reviewers had concerns about the novelty of the approach. Including being confused about whether this is just an application of curriculum learning. Reviewers were also concerned about the lack of ablations.

Reviewers also had concerns about the fact that this is a toy domain. Symbolic mapping as defined in the manuscript appears to be possible only for such toy domains. It fundamentally wouldn't scale to simple language games with real images. This significantly limits the scope of the work. More broadly, reviewers wanted to see symbolic mapping exercised much more. If this is a useful idea, they wanted to see the authors apply it to other domains.

Reviewers were confused about many other details in the manuscript. For example, about the fact that refdis is later discarded as a metric, which the authors answered is due to redundant symbols ("the symbolic mapping is not a highly compositional representation here because of the redundant symbols"). Why redundant symbols lead to less compositional representations seem unclear.

With significant additional improvements to the clarity of the manuscript, a demonstration of how symbolic mapping is useful in another domain, and additional experiments suggested by multiple reviewers this could be a strong submission in the future.